# Muscle Diversity, Heterogeneity, and Gradients: Learning from Sarcoglycanopathies

**DOI:** 10.3390/ijms22052502

**Published:** 2021-03-02

**Authors:** Carles Sánchez Riera, Biliana Lozanoska-Ochser, Stefano Testa, Ersilia Fornetti, Marina Bouché, Luca Madaro

**Affiliations:** 1Department of AHFMO, University of Rome “la Sapienza”, Via A. Scarpa 14, 00161 Rome, Italy; biliana.lozanoska-ochser@uniroma1.it (B.L.-O.); marina.bouche@uniroma1.it (M.B.); luca.madaro@uniroma1.it (L.M.); 2Laboratory of Compared Anatomy, Department of Biology, University of Rome Tor Vergata, Via della Ricerca Scientifica s/n, 00133 Rome, Italy; stefanotesta87@hotmail.it (S.T.); ersiforn@hotmail.it (E.F.)

**Keywords:** sarcoglycanopathies, muscle heterogeneity, muscle diversity, gradients, new technologies

## Abstract

Skeletal muscle, the most abundant tissue in the body, is heterogeneous. This heterogeneity forms the basis of muscle diversity, which is reflected in the specialized functions of muscles in different parts of the body. However, these different parts are not always clearly delimitated, and this often gives rise to gradients within the same muscle and even across the body. During the last decade, several studies on muscular disorders both in mice and in humans have observed particular distribution patterns of muscle weakness during disease, indicating that the same mutation can affect muscles differently. Moreover, these phenotypical differences reveal gradients of severity, existing alongside other architectural gradients. These two factors are especially prominent in sarcoglycanopathies. Nevertheless, very little is known about the mechanism(s) driving the phenotypic diversity of the muscles affected by these diseases. Here, we will review the available literature on sarcoglycanopathies, focusing on phenotypic differences among affected muscles and gradients, characterization techniques, molecular signatures, and cell population heterogeneity, highlighting the possibilities opened up by new technologies. This review aims to revive research interest in the diverse disease phenotype affecting different muscles, in order to pave the way for new therapeutic interventions.

## 1. Introduction

Skeletal muscle comprises almost 40% of the weight of the body and under normal conditions can perform multifunctional tasks such as muscle contraction, temperature regulation, energy storing, skeleton protection, and homeostasis maintenance [1,2,3]. To achieve this the tissue relies on multinucleated contractile muscle cells, known as myofibers [4]. Myofibers, together with motor neurons, form the muscle functional subunits, which are repetitive structures with similar functional properties [5]. During development, skeletal muscle growth is sustained through the fusion of muscle precursor cells. During adulthood, the muscle stem cells, called satellite cells (SCs), remain in a quiescent state beneath the basal lamina, being activated sporadically to guarantee the physiological turnover caused by daily life [4]. Despite the fact that all muscles are characterized by almost identical structures, similar composition, and capacity for regeneration, skeletal muscle is very far from being a homogeneous tissue.

Numerous studies have tried to address different aspects of muscle heterogeneity, including muscle fiber specialization [6,7], excitability [8,9], metabolism [10], molecular pathways [11,12], contraction mechanism [13], and calcium kinetics [14]. This heterogeneity, seen across each muscle, can be understood by studying the plasticity that allows them to exert different tasks such as low-intensity activity (e.g., posture), repeated submaximal contractions (for example, locomotion), or fast and strong maximal contractions (jumping, kicking) [15]. However, even if muscles have a great capacity to adapt to changing conditions based on their heterogeneity, not all of them can perform the same functions. Muscles designed for fast and strong maximal contractions such as triceps cannot become a postural muscle (characterized by low-intensity activity). Thus, triceps can improve their endurance after a few weeks of training, but the improvement will not be linear and eventually will reach a plateau [16]. These limitations are not due to just muscle heterogeneity, given the different array of the affected muscle fibers, but also due to muscle diversity, whereby each muscle has its own specific morpho-functional characteristics.

Until now, muscle diversity has been associated with species, gender, and individual polymorphisms leading to muscle diversification during development [15]. However, there are still several important issues to address, in order to better understand the physiology and pathophysiology of the muscle: for example, why during disease muscles carrying the same genetic mutation degenerate at a different pace.

Indeed, differences in disease severity within different muscles can be observed in most of the muscular dystrophies such as Duchenne muscular dystrophy [17], desminopathy [18], calpainopathy [19], and dysferlinopathy [20] (Figure 1). However, in the sarcoglycanopathies, particularly a group of the limb-girdle muscular dystrophies (LGMDs), these differences are particularly pronounced [21,22,23,24,25]. Interestingly, in all these pathologies there is a phenotypic gradient based on the degree of disease severity affecting different parts of the musculature. Thus, the appearance of gradients and their “flow” among different muscle structures may form the basis of muscle diversity.

During recent years, advances in technological analyses have enabled the characterization of the muscles’ specialized profiles [26]. Specifically, transcriptomics and single cell analysis could allow a remarkably detailed characterization of diverse muscles in order to understand why some muscles resist degeneration.

In this review, we will focus our attention on the differences among muscles of the same organism. We will give an overview of the published data on muscle differences, characterization techniques, molecular signatures, and cell populations, discussing the possibilities opened up by new technologies. We will also discuss the potential of using our new understanding of muscle disease diversity for future therapeutic interventions.

## 2. Sarcoglycanopathies

Within the LGMDs, sarcoglycanopathies show particularly prominent differences among muscles. This group results from mutations in one of the four genes encoding for the transmembrane proteins of the sarcoglycan (SG) complex (α-, γ-, δ-, and β-sarcoglycan) [22,27]. All sarcoglycanopathies are autosomal recessive disorders and are considered rare diseases as the worldwide reported frequency is around 1/200,000–1/350,000 [28,29]. Additionally, two other sarcoglycans, SGCE and SGCZ, are present in the smooth muscle SG complex, but mutations in their genes have not been linked to any skeletal muscle disease yet [25]. The formation of the SG complex begins in the endoplasmic reticulum with the recruitment of the beta-sarcoglycan subunit [30], followed by the gamma, delta, and alpha subunit incorporation into the complex. Importantly, the loss of any of these proteins leads to a deficiency of the whole complex [31].

The role of the SG complex has not been fully elucidated [31,32,33]. As part of the dystrophin-associated glycoprotein complex (DAGC) that attaches the cytoskeleton to the extracellular matrix, the complex preserves the membrane stability against contraction forces [34,35] (Figure 2). Moreover, it might participate in unknown molecular pathways involved in muscle maintenance, since the loss of sarcoglycans results in muscle degeneration even without mechanical injury [36,37]. Furthermore, the SG complex is expressed in both the nervous and muscular system; yet it is only the latter that is affected in these diseases [38].

Clinically, sarcoglycanopathy patients present common traits such as elevated serum levels of creatine kinase (CK), variable muscle weaknesses (predominantly in the proximal muscles), calf hypertrophy, difficulties in arising from the floor, and the progressive loss of ambulation [23]. In the last stages of these diseases, up to 25% of patients need ventilator support [25] and cardiac involvement is present in as many as fifty percent of the cases, being more frequent in beta and gamma-sarcoglycanopathy [39,40]. However, despite their common traits, patients with sarcoglycanopathies have a very heterogeneous phenotypic profile ranging from severe Duchenne-like phenotype to mild myopathic changes [22,24,41,42] suggesting a weak correlation between genotype and phenotype. Indeed, within a large cohort of Chinese patients, a correct prediction of genotype based on the expression of sarcoglycan proteins was achieved in only 36.0% of patients [43]. Conversely, another study in alpha-sarcoglycan patients described a correlation between the residual expression of this protein and the severity of the disease [25]. No correlation for beta or gamma-sarcoglycanopathy patients could be found in the same study even though a combination of risk factors associated with rapid disease progression in these patients was identified. The low predictive power of gene mutations suggests that, in addition to gene mutations, phenotypic heterogeneity likely results from a range of other indirect and so far unknown factors.

Currently, different strategies have been proposed to prevent muscle degeneration in sarcoglycanopathies, including cell therapy [44], anti-inflammatory treatments [45], or gene therapy [46,47]; however, there is still no available cure.

## 3. Contractile and Non-Contractile Tissue Heterogeneity

Skeletal muscle is composed of contractile and non-contractile tissues. The contractile component, which is the most studied and relevant in terms of mass, consists of a variety of functionally diverse muscle fiber types [15]. One simplified classification based on their contraction and metabolic characteristics includes slow twitch oxidative and fast twitch glycolytic fibers. This fiber type classification distinguishes four main types that can be easily identified using histochemical staining techniques based on metabolic enzyme activity, such as succinate dehydrogenase (SDH) or nicotinamide adenine dinucleotide (NADH) [48]. The relative abundance of any of these fiber types within the muscle may vary between species and anatomical sites giving place to muscle heterogeneity. This heterogeneity depends on the myosin variant expressed, which might result in contractile activity features. As many as 16 different sarcomeric myosin isoforms have been described so far [15]. Some of these myosins are exclusively expressed in certain muscles. For example, jaw muscles express myosin heavy chain 16 (MYH16) that cannot be found in the trunk and limb skeletal muscles. Another source of heterogeneity can be of embryological origin. Interestingly, head muscles do not derive from somites as trunk and limb muscles do, but from presomitic cranial mesoderm [49]. However, muscle fiber heterogeneity is not restricted to myofibrillar proteins [50], metabolic enzymes [48], or developmental origin [51,52,53], but it involves also subcellular systems, including transmembrane ionic fluxes and intracellular calcium signaling [15]. These aspects require more detailed study to further understand the varying degree of disease severity among different muscles in sarcoglycanopathies.

In addition to the contractile muscle fibers, a large variety of non-contractile components contribute to the muscle organization and function, and might also contribute to muscle diversity. Among these, the connective tissues, forming tendons, ligaments, the three fascia layers (endo-, peri-, and epimysium), and the extracellular matrix (ECM) in general strongly influence muscle function [54]. Nonetheless, after acute muscle injury, transient extracellular matrix (ECM) remodeling is essential for normal muscle repair [55]. During disease, an aberrant ECM remodeling process leads to the perpetual stiffening of the ECM and ultimately creates a barrier that impairs skeletal muscle regeneration, giving rise to the most typical muscular dystrophy hallmarks: fibrosis and fat deposition [56].

Collagen is the major structural protein of these non-contractile components, accounting for 1–10% of muscle dry weight [57]. At least 28 isoforms of collagen exist across the body; among them fibrillar types I and III predominate in the adult endo-, peri-, and epimysium [58]. It is interesting that the perimysium and tendon both contain primarily type I collagen, and decorin, as the most abundant proteoglycan in both structures [54]. In contrast, epimysium and endomysium contain almost equal amounts of types I and III collagen and proteoglicans other than decorin. This fact supports the hypothesis that the perimysium is in continuity with tendons and that is very relevant due to two different factor. First, that the perimysium is the space where ECM changes mostly take place during muscle repair and during disease [59], and second, that differences among muscle-tendon arrangements have been observed along the body. For example, short fascicles are associated with long thin tendons promoting elastic energy saving, whereas long fascicles are associated with little or no tendons, favoring the ability of a muscle to produce mechanical power and control length and position [60]. These observations suggest that tendons, and the perimysium, may play a role in the development of diverse muscle phenotypes, which could have direct implications in how we approach sarcoglycanopathies.

Thus, even though the structure and function of connective tissues have been studied for some muscles [61,62], little is known about the heterogeneity within these non-contractile tissues, and even less about the morpho-functional features within the transition areas (e.g., from tendon to perimysium), especially because of the difficulty associated with isolating these “regions”.

Indeed, it is likely that some degree of diversity of connective tissues exists across the body, and it may have a potential role in muscle disease modulation.

## 4. Cell Population Heterogeneity

Another factor that may contribute to muscle diversity is the heterogeneity of cell populations within the muscle. Indeed, muscle homeostasis is maintained through the cooperative actions of a variety of cell populations [4]. Among these populations, muscle stem cells, otherwise known as satellite cells [63], are responsible for myofiber growth upon activation and fusion. In addition to satellite cells, other mesenchymal stem cells found within the muscle, such as adult bone marrow derived cells [64], fibroadipogenic progenitors (FAPs) [65], mesoangioblasts [66], pericytes [67], and PW1^+^ interstitial cells (PICs) [68] can contribute to muscle growth and regeneration. Furthermore, during these processes, the activity of stem cells is supported by various resident and infiltrating immune cells [59,69], which help the fibers to regenerate following acute damage or in preparing the local environment for the growth and deposition of ECM during disease. Indeed, the activity of the immune system is necessary to counteract muscle degeneration in a mouse model of LGMD-R4 [70]. Another important issue to be considered for understanding these interactions in sarcoglycanopathies, as well as in other muscular dystrophies, is the regulation of the migration abilities of the different cell populations which might represent an important modulation factor [71,72].

Previous studies have described the heterogeneity of some stem cell populations, especially satellite cells. The heterogeneity of satellite cells has been described in [73], and is reflected by gene expression profiles or cell surface markers [74,75,76], as well as by their differentiation capabilities [77,78]. Likewise, other interstitial cell populations such as FAPs, isolated in mouse as lin^−^ Sca1^+^, showed different levels of specific markers, such as Vcam1 and Tie2, depending on the context of the damage; acute or chronic injury [79]. Thus, single-cell gene expression profiling of skeletal muscle-derived cells has emerged as a very useful tool [80]. In addition, the work by Giordani and colleagues [81] took advantage of mass cytometry and transcriptomic profiling to reveal understudied muscle cell populations, thereby adding further complexity and highlighting the importance of uncovering new (sub)populations. Similarly, in the context of sarcoglycanopathies, Camps and colleagues [82] used single-cell technology to reveal, once again, an extensive heterogeneity among interstitial muscle cells. In particular, this study reports differences in adipogenic interstitial cells directly related to dystrophic muscles.

These interesting discoveries have raised the question as to whether additional cell subpopulations or different arrays of the already identified cell populations might be involved in muscle diversity and gradient. Indeed, we wonder whether an eventual gradient distribution of these cells may be conditioned by this context. Hence, since not all muscles respond equally to the same mutation, we believe that mouse models of sarcoglycanopathies might represent an extremely important source of different contexts in which to investigate cell population heterogeneity and their network interactions.

## 5. Gradients and Sarcoglycanopathies

Muscles are apparently well-defined structures delimitated by their connective layers. However, due to their heterogeneity, it is likely that they are characterized by a gradient flow rather than as homogeneous single structures. In sarcoglycanopathies, these well-defined structures were clearly described in the work of Tasca and colleagues [21]. They showed unequivocally that muscles behaving totally differently coexist in the same leg compartment. In fact, spared or even hypertrophied muscles such as gracilis, sartorious, and rectus femoris can be observed, in proximity to the degenerating thigh adductors and posterior thigh muscles, which are the most severely affected muscles in these diseases (Figure 1).

Nonetheless, in the same study, considering the whole-body tissue musculature, these diverse muscles that seem to work as delimitated defined structures showed an evident proximo-distal gradient [21], recognized by the gradual change in disease severity from proximal to distal parts. This idea was supported by the observation that the distal leg compartment displays a normal or relatively preserved MRI, whereas the proximal limb muscles show a degenerated pattern. Indeed, this gradient was observed also within the same muscles, such as the quadriceps, where the distal part was spared while the proximal part degenerated.

Interestingly, the idea of a gradient along the musculature becomes more apparent when the body is studied from an architectural and geometrical point of view [60,83,84,85]. Previous studies have demonstrated a proximo-distal gradient in limb muscles as the reflection of the best compromise between energy saving, function, and architecture. Proximal limb muscles, such as the hip extensors, have long, parallel fibers that provide mechanical power, fast contraction, and precise control of joint positions to overcome gravity and move the body’s center of mass during locomotion. On the other hand, muscles with smaller mass, shorter fibers, higher pennation angles, and long compliant tendons are generally located distally within terrestrial limbs [60,83]. The collagen concentration also contributes to the above gradient, since the proximal part of the limbs contains less collagen than the distal [84].

In sarcoglycanopathies, the existence of these proximo-distal gradients reflected both in muscle pathological features and in the musculo-tendinous architecture suggest a possible unexplored relationship between the non-contractile elements such as the tendon-perimysium and muscle resistance (Figure 3).

## 6. Muscle Diversity: Evidence from Murine Models

Muscle heterogeneity, as well as many other aspects of muscle physiology, have been widely studied in mouse models (see Table 1) [86,87]. However, mouse models do not always reflect the human dystrophic outcome due to mutations.

**Table 1 ijms-22-02502-t001:** Summary of sarcoglycanopathies mouse models.

Name	Mutation	Target Sarcoglycan	References	Phenotype	Heart Involvement
Alpha-Sarcoglycan Knock Out	Missense mutation (H77C) exon 3	Alpha	Kobuke et al. 2008 [88]	Progressive muscle degeneration	No
Alpha-Sarcoglycan Knock Out	Sgca (H77C) + transgene human Sgce	Alpha	Kobayashi et al. 2008 [89]	No dystrophic symptoms but fatigue	No
Alpha-Sarcoglycan Knock Out	Deletion Exon 2–3	Alpha	Duclos et al. 1998 [90]	Progressive muscle degeneration	No
Alpha-Sarcoglycan Knock Out	Deletion Exon 1–2	Alpha	Liu et al. 1999 [91]	Progressive muscle degeneration	No
Alpha-Sarcoglycan Knock Out Immuno Deficient	Sgca/Rag1/il2rg Knock out	Alpha	Duclos et al. 1998 [90] Mombaerts et al. 1992 [92] Cao et al. 1995 [93]	Progressive muscle degeneration	No
Beta-Sarcoglycan Knock Out	Deletion exon 3–6	Beta	Durbeej et al. 2000 [94]	Progressive muscle degeneration	Yes
Beta-Sarcoglycan Knock Out	Disruption exon 2	Beta	Araishi et al. 1999 [95]	Muscle degeneration and hind limb hypertrophy	Yes—Old age
Beta-Sarcoglycan Knock Out	Sgcb/Rag2/γc Knock out	Beta	Giovannelli et al. [70]	Exacerbated dystrophic phenotype	Yes
Beta-Sarcoglycan Knock In	Missense mutation T153R KI Exon 4	Beta	Henriques et al. 2018 [96]	No symptoms	No
Gamma-Sarcoglycan Knock Out	Deletion exon 2	Gamma	Hack AA et al. 1998 [97]	Progressive muscle degeneration	Yes
Gamma-Sarcoglycan Knock Out	521ΔT Single nucleotide deletion Exon 6	Gamma	Demonbreun et al. 2020 [98]	Progressive muscle degeneration	Yes

Among the approaches that have been used to highlight the differences among muscles, it is important to mention the study by Straub and colleagues [99]. In 1997, they looked for differences between congenital muscular dystrophy and Duchenne muscular dystrophy, since the genetic mutations in these two dystrophies affected components of the same complex (dystrophin-associated glycoprotein complex), and yet showed different levels of severity. They injected Evans blue dye into the blood stream of mice. This dye can enter into skeletal muscle fibers when the sarcolemma is disrupted, as happens in these dystrophies. Interestingly, they observed that some muscles, but not all of them, were blue colored in both mouse models, but with different disease-specific patterns. They also showed that some muscles were resistant to sarcolemma disruption in a disease-specific pattern, demonstrating that in muscular dystrophies muscles differ in how severely they are affected by the same mutation.

In this regard, another study to be mentioned is the one by Sasaoka et al. [100], where they focused on understanding the hypertrophy phenomenon taking place in gamma-sarcoglycanopathy and other dystrophies. To this aim, they took the entire lower part of the hind limb and cut transversal sections, upon decalcification, to observe all the muscles of the leg together in the same section. They showed clear differences in fiber structure and cross-sectional areas among muscles within the same hind limb section.

The hypertrophy of some muscles is a common feature in the sarcoglycanopathies, but also in other dystrophies such as Duchenne muscular dystrophy, with which it shares remarkably similar gene expression profiles, including inflammatory and structural remodeling processes [86,100,101]. The MRI technique has been used to show hind limb muscle differences in humans [21], however, due to the small size of mouse muscles it is difficult to interpret the results. Hence, the use of MRI techniques to study muscle differences in mouse models has not been extensively reported upon [102,103].

On other hand, in recent years, studies of gene expression profiles have been particularly successful in exploring muscle diversity and heterogeneity. Transcriptomic profiles have been used to study fiber types [104], confirming that differences in fiber types are likely regulated by multiple signaling pathways and transcription factors rather than by a single “master” switch [105]. Another study of transcriptomic profiles evaluated the differences among wild type muscles and found more than 50% differentially expressed transcripts [26].

## 7. Conclusions and Future Perspectives

Skeletal muscle heterogeneity enables multitask functions and over the past century has been the focus of numerous studies and efforts to characterize it. This heterogeneity encompasses a broad range of characteristics, including fiber type, embryogenic origin, metabolism, calcium kinetics, architecture, excitability, and cell populations. However, not all muscles of the body can accomplish all the functions that the whole coordinated musculature can perform, and this forms the basis of muscle diversity.

Muscle diversity is reminiscent of that in the skin. Thus, in the same way the skin differs between the palm of the hand and that on the face, the muscle heterogeneity is reflected in the distinct parts of the body through its diversity. Furthermore, diversity occurring along different tissues may determine and influence a specific area.

A suitable example of that diversity could be seen in sarcoglycanopathies, where the pelvic muscles and shoulder limbs are known to be preferentially affected. Even though all muscles in these pathologies have the same genetic mutation, not all of them react equally to the same genetic background, as is also the case in other muscular dystrophies. Considering also that muscle degeneration occurs even without physical activity in these diseases, it is conceivable that a variety of unknown molecular pathways and cell (sub)populations are taking part in mediating the different muscle phenotypes and behaviors. Therefore, we believe that the current classification of muscles, which does not take into account this phenomenon, is not exhaustive, and it should be revised in the near future when more molecular pathways and cell populations are unveiled. Understanding the complexity of muscular diversity will no doubt allow the development of novel ways to interfere with muscular diseases.

Furthermore, the authors of this review believe that further studies are required to characterize the non-contractile (connective) tissues, as part of the muscles. Indeed, the ECM, one of the components of the muscle and the place where the most important changes during muscle pathology occurs, deserves to be further explored. In the context of muscular diseases and, especially in sarcoglycanopathies, a proximo-distal gradient along the muscles can be observed. The most severely affected muscles of this gradient have as a hallmark a huge accumulation of ECM (fibrosis). On the other hand, muscles contributing to the proximo-distal gradient within sarcoglycanopathies are interestingly also part of another proximo-distal gradient: an architectural one. Proximal muscles have longer fibers, shorter tendons, and less collagen (in terms of relative mass) than distal muscles, which have shorter fibers, bigger tendons, and more collagen. Hence, it is plausible that some of the unknown molecular pathways of muscular resistance observed in sarcoglycanopathies involve the ECM and its diversity.

In order to address these outstanding issues it is therefore necessary to improve muscle characterization studies by taking advantage of new technologies such as RNA bulk and single cell sequencing which should help in understanding the borders of these gradients, the muscle tissue diversity, as well as the principal actors in these complex and diverse muscular niches.

## Figures and Tables

**Figure 1 ijms-22-02502-f001:**
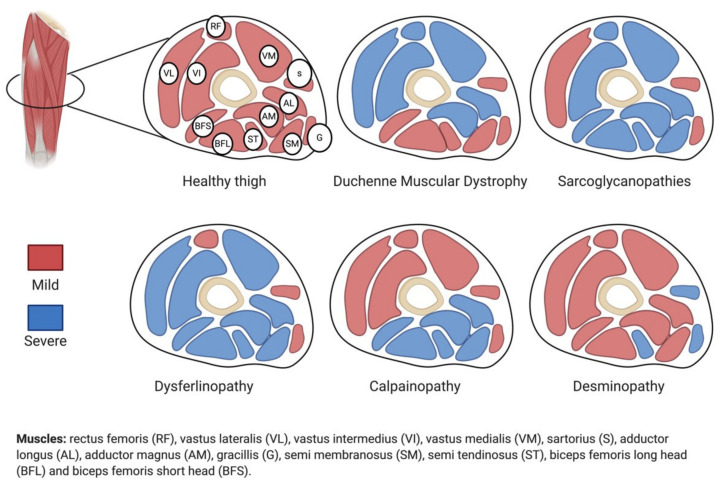
Schematic presentation of thigh human transversal sections: level of damage severity of the muscle groups in different muscular dystrophies. MRI-based representation of human thigh (middle) transversal sections. Red color: mild muscle pathology or late disease stage; blue color: severe muscle pathology or early stage of the disease.

**Figure 2 ijms-22-02502-f002:**
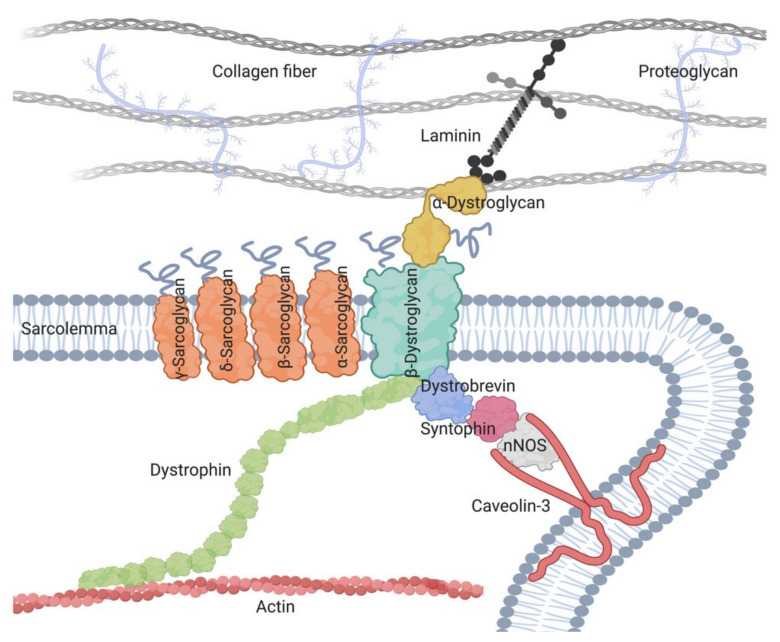
A schematic presentation of the dystrophin-associated glycoprotein complex (DAGC), including its main components and its localization around cell sarcolemma.

**Figure 3 ijms-22-02502-f003:**
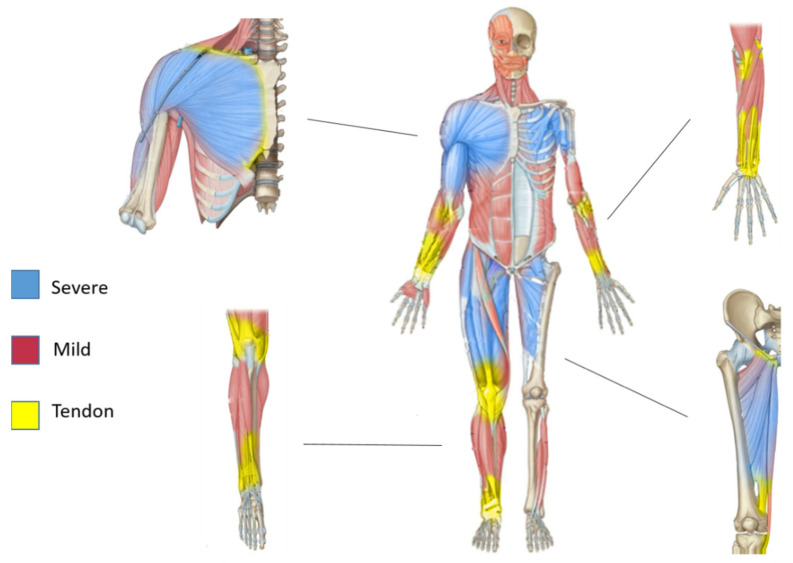
Proximo-distal pattern of muscles affected in sarcoglycanopathies and proximo-distal pattern in tendon architecture. Schematic presentation of human sarcoglycanopathy-affected muscle pattern with the magnification of Scheme.

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
