# Peer review of "Muscle Diversity, Heterogeneity, and Gradients: Learning from Sarcoglycanopathies"

_ijms, 2021, doi:10.3390/ijms22052502_

Round 1

Reviewer 1 Report

The flow and connectivity of the contents in the manuscript is indeed not excellent.

To emphasize the importance of the article, the abstract should be revised in a more precise and structured way.

other seems to be fine 

The authors have made a good effort to improve the paper, and now it's in better condition.  In fact, the flow and connectivity of the contents in the manuscript are not excellent. In order to emphasize the importance of the article, the abstract should be revised in an even more precise and structured manner. generally, a review article should include/review recent findings in the field, but here recent references are less.   My recommendation for this paper is for 'minor revision'

Author Response

  Response to Reviewer 1 Comments
Point 1: The authors have made a good effort to improve the paper, and now it's in better condition. In fact, the flow and connectivity of the contents in the manuscript are not excellent. In order to emphasize the importance of the article, the abstract should be revised in an even more precise and structured manner. generally, a review article should include/review recent findings in the field, but here recent references are less. My recommendation for this paper is
for 'minor revision'.

Response 1: We appreciate the comments, we believe they help us to improve the review a lot. Accordingly to that, we have rewrite the abstract in a more precise manner. Thanks. 

Reviewer 2 Report

This paper has now been much improved. However, there are some remaining issues that need to be addressed. The major complaint is the definition of the gradient. The gradient appears in the beginning (line 63) undefined, and then is discussed in section 5, still without a proper definition. Please add how the gradient is defined.

Minor comments:

Line 175, define PIC

Author Response

Response to Reviewer 2 Comments

Point 1: This paper has now been much improved. However, there are some remaining issues that need to be addressed. The major complaint is the definition of the gradient. The gradient appears in the beginning (line 63) undefined, and then is discussed in section 5, still without a proper definition. Please add how the gradient is defined.

Response 1: We thanks the reviewer for the comments. We change the abstract explaining better the concept of gradient and we also change some sentence from line 63 and section 5. We hope now it is more clear the definition of the concept. Thanks.

Minor comments:

Line 175, define PIC

Response 2: We change it. Thanks.

This manuscript is a resubmission of an earlier submission. The following is a list of the peer review reports and author responses from that submission.

Round 1

Reviewer 1 Report

This reviewer expected to read how muscle diversity and heterogeneity, as well as a gradient, contribute to sarcoglycanopathy but found just a very vague and disconnected description of muscle heterogeneity, diversity, gradient, and sarcoglycanopathies, as well as a highly speculative suggestion that the sarcoglycanopathies diverse phenotype is the consequence of muscle heterogeneity, diversity, and gradient. Muscle heterogeneity, diversity, and gradient are defined poorly, especially muscle heterogeneity and diversity. The reader may have an idea that the authors do not understand the difference between these terms by themselves. The central part of the manuscript, describing sarcoglycanopathies is a poor compilation of the Wikipedia page, a couple of sentences from the abstracts of a few cited papers, and some references from the Pubmed search. No analysis of the referenced results is provided for a reader. The purpose of Table 1 is not clear. Table 1 referred in the text just once and in a sort of negative context, saying that the mouse models do not necessarily reflect the case of dystrophy in humans. Why bother to report those results then? Some English editing is necessary.

Reviewer 2 Report

  1. Figure 1: Correct the spelling “Healty Thigh” it should be ‘healthy thigh’
  2. Delete “The represented muscles…. short head (BFS).” From the caption of figure 1 or crop the same from the image (figure1)
  3. Authors should combine the two small paragraphs and make them concise. For example, from lines number 83-101, there are four paragraphs, which should be a maximum of 2.
  4. Redraw Figure 2 in a more precise and detailed way and revise (rewrite) its caption.
  5. The authors should define the meaning of ‘KO’ in table 1.
  6. Future perspectives and conclusion should be included
  7. If possible, the molecular mechanism of myogenesis and the extracellular matrix may be discussed with a focus on myopathies.